

# Assessment of Uranium concentration in groundwater and its human health impact in a part of Northern Tamil Nadu, India

Sundaram Parimalarenganayaki *[1], Email*: parimala.renganayaki@vit.ac.in

Alagarsamy Rahul [1], arahul.2016@vitstudent.ac.in

Mahamad Hussen Thabrez [1], thabrez.m@vit.ac.in

Rajendran Anbuchezhian [1], anbuchezhian.r@vit.ac.in

Subramanian Manoj [2], leaveittomanoj@gmail.com

Lakhsmanan Elango [2], elango34@hotmail.com

[1] School of Civil Engineering, Department of Environment and Water Resources Engineering, Vellore Institute of Technology, Vellore,630104,India
[2] Department of Geology, Anna University, Chennai -600 025, India

**Abstract**. Ground water is the major source of drinking water in India. Over exploitation of this water resource has exacerbated the situation of providing good quality water, due to the presence of major ions, minor ions, trace elements and radioactive elements. A study has carried out in Vellore, a city in the Northern part of the Tamilnadu, to understand the levels of Uranium concentrations and other geochemical parameters in drinking water. Samples were collected and the analysed for various water quality parameters in the laboratory and as well as insitu. Uranium concentration was estimated using the Quantalase Laser Fluorimeter and nearly 10 % of the samples show that the concentration of Uranium is exceeding the permissible limit .The low observed adverse effect level and No observed adverse effect level values were also assessed for to understand the risk due to the presence of the uranium in drinking water and it was observed that there is a slight risk on the public health due to the consumption of ground water in this region.

**Keywords**: Groundwater quality; Vellore; Radioactive elements

## 1. Introduction

Uranium (U) occurs on earth by natural means. Minute quantities of uranium can be found in humans, animals, plants, soil, water and rocks. Even though it is radioactive, it exhibits weak radioactivity and adds to the low levels of radiation occurring naturally in the environment. Since U is present mostly everywhere in earth, it is also present in significant quantities in water, especially in drinking water. Due to natural erosion and weathering process, uranium mobilized from the rocks to groundwater and surface water. Some of the methods by which U can get into drinking water are human activities and geogenic sources such as mining and dissolution of U containing minerals in groundwater respectively (Bruce et al., 2014). Mainly U is present in groundwater rather than surface water, so there is a need to assess the amount of U present in groundwater.



Higher amounts of U content can affect some parts of the body (Brugge and Oldmixon, 2005). Higher U content
can affect the kidneys and it is caused due to its chemical nature and not due to radioactive property (Kurttio et
al., 2006). In correlation to the previous statement, several studies were carried out across various countries to
assess the U content in groundwater. Anderson et.al (2003) reported from Colorado River, USA that U
concentrations in the groundwater within the test area range from 0.4 to 1.4 μM and was above the U Mill
Tailings Remedial Action (UMTRA) maximum contaminant limit of 0.18 μM. Nolan et al (2015) reported that
the High Plains and Central Valley aquifers in USA, two of the largest and most productive aquifers in the
world, are among aquifers with high concentrations of dissolved U in groundwater. Orloff et al (2004) reported
high concentrations of U (mean = 620 μgl$^{-1}$) in water samples collected from private wells in a residential
community in USA. In India the contamination of water due to was first detected in India in the year 2009, in a
research of heavy metal toxicity. High levels of U were found in 88 % of the samples; the levels were more than
60 times the maximum safe limit of 30 ppb as prescribed by World Health Organisation (WHO, 2011). Brindha
et al. (2013) reported high U concentrations from 0.2 to 118.4 ppb in groundwater in a part of Nalgonda district,
Andhra Pradesh, India. Kumar et al (2015) obtained results that showed variations in concentrations obtained
from place to place and values ranged from 11±0.76 to 63.33±2.28 μgl$^{-1}$ of U in the study of ground water at
random locations of Varanasi, Allahabad, Kaushambi and Fatehpur districts in Uttar Pradesh, India. Rana et al
(2015) studied the ground water around a newly established U mining and processing facility at Tummalapalle,
Vemula Mandal of Kadapa district in the state of Andhra Pradesh, India. The U concentration in groundwater
samples was found to vary between 0.38 and 79.70 ppb. Babu et al (2008) reported the groundwater U
concentration range 0.3-1442.9 ppb in Kolar district, South India. The results of a study from Bajwa et al (2017)
showed that a large variation (0.5–579 μgl$^{-1}$) in the uranium concentrations in drinking water of the South west
parts of Punjab.

Tamilnadu, a state in the South India, where bore wells and hand pumps forms a major source of drinking water
for the people living in the region. The primary reason is that rainfall in Tamilnadu is highly seasonal and during
these recent years, the rainfall has drastically been affected due to various human factors. Thus, it is very
important to provide clean and safe drinking water to the people as they depend on it at most. The aim of the
present study was to find the U concentration in the groundwater in Vellore city of Northern Tamilnadu.

**2. Study area**
The present study was conducted in regions of the Vellore (12.9165° N, 79.1325° E) a city in Tamilnadu, India.
The city is located 220 m above mean sea level with a semi-arid climate. It lies near the Eastern Ghats and the
Palar river flows across the city which is non-perennial. The average annual rainfall of the city is 950 mm.
September being the wettest month, the southwest monsoon brings majority of the rainfall to Vellore. The
hottest months are April to June (maximum of 39.4° C) and coldest months are December-January (minimum of
18.4° C). Most of the rivers in the district are dry and the rainfall pattern has been erratic so this has resulted in
the over exploitation of ground water through open wells and bore wells. The relative humidity in Vellore lies in
the range of 37-85 %. Soil types of this study area are loamy and clayey, whereas geological classification may
include alluvium, granite, gneisses and charnockite. Erratic rainfall pattern has resulted in the over exploitation
of groundwater.



**3. Methodology**

**3.1. Sampling and analysis**

Study area was divided into 2 km by 2 km grid in order to ensure the uniform distribution of sample locations and later, the coordinates of the centre of each grid were identified with the help of satellite images. The groundwater samples were collected from hand pumps and bore wells at these locations within a radius of 5 km during the month of September 2017 (Fig. 1). The parameters such as Electrical conductivity, pH, and Dissolved oxygen were analysed in situ using YSI digital multi-parameter instrument kit whereas the Alkalinity and Hardness were measured by the titration method in the laboratory.

The concentration of U in the groundwater samples was estimated using Quantalase LF2a Laser Fluorimeter. The instrument was calibrated in the range of 1 to 100 ppb using a standard stock solution and the Phosphoric acid in ultra-pure water was used as fluorescence reagent in the analysis. To obtain blank counting, a blank sample with same amount of fluorescing reagent was measured for U concentration.

**4. Results and Discussion**

The concentration of uranium and other physiochemical parameters at different locations in the study area are tabulated in the Table 1. Weathering and dissolution of rocks and soils constitute TDS naturally into groundwater. In the present study the minimum and maximum values of TDS varied from 328 µgl$^{-1}$ to 4671 µgl$^{-1}$. TDS of groundwater in this study area is very high when compared with standards such as WHO and SEPA. Similar to TDS, the values for the EC ranging from 489 µScm$^{-1}$ to 6971 µScm$^{-1}$. As this area is mostly covered by hard rocks, the samples results show the high TDS and EC values. 84 % of groundwater samples are exceeding the limit i.e. 500 mgl$^{-1}$. The pH values are lies between 6.1 and 7.85. Whereas the minimum and maximum ranges of DO vary from and 3.8 mgl$^{-1}$ to 7.3 mgl$^{-1}$ respectively. The mean, maximum and minimum values of this data are given in the Table 2.

Figure 2 shows the relationship between percentage U and its concentration. It is observed that maximum of 62.26 % of samples in the study area has U concentration of 0-10 ppb while 16.98 %, 7.55 %, 13.2 % of samples are in the range of 11-20, 21-30, >30 ppb respectively.

The minimum U concentration value observed was 0.3 ppb and the maximum value observed was 69.5 ppb with an average value of 12.94 ppb. The permissible limits of the uranium concentration in the drinking water across the different bodies of the world are given in the Table 3. As per the AERB the guideline limit is 60 ppb, in this study at only one location it has crossed this prescribed limit. As per, WHO and USEPA the permissible limit of U concentration in drinking water is 30 ppb and it is been observed that out of the 53 ground water samples, in 7 samples the concentration of Uranium is exceeding this prescribed limit (i.e. >30ppb). Spatial distribution of U in the study area is shown in Fig. 3. The high concentration of U in Northern side is may be due to leaching of charnokite, gneiss and granite present in that region. The correlation between U and other parameters are shown in Fig. 4. It is observed that there is no strong correlation of U with pH, Dissolved Oxygen and alkalinity but a slight positive correlation is observed between the U and TDS. Coefficient of correlation of the Uranium concentration with the various parameters in drinking water is given in Table 4. This implies that 'U' may be present in water due to the increased presence of dissolved solids. Since there is no strong correlation with these parameters it implies that the presence natural uranium in this area might be due to geological formations and other factors.





**5. Human health impact of Uranium**
Uranium is a radionuclide that emits primarily alpha particles and is associated with many health risks. Uranium
is a health hazard only if it is taken in to the body as it is an alpha emitter. The Uranium contaminated water
does not cause any radiological effects although chemically it can affect the human body. Kidneys are the
primary targets of U contamination. A higher Uranium trace causes the failure of the functioning of the kidneys.
Oesophagus and stomach cancers are also an effect of continuous consumption of U contaminated water.
**5.1. NOAEL/LOAEL**
Various researches have been carried out in order to understand the toxicological effect of U in drinking water.
No observed adverse effect level (NOAEL) is the highest toxic point at which there is no adverse effect to any
human due to the toxicity, whereas, low observed adverse effect level (LOAEL)  is the lowest toxic point at
which there is adverse effect occurs due to the toxicity. LOAEL is slightly higher than NOAEL in one dosage.
According to Public Health Goal (PHG 2001) NOAEL or LOAEL is given as following Equation

$$C = \frac{(NOAEL/LOAEL) \times BW \times RSC}{UF \times W}$$

C = Public health-protective concentration for U in drinking water ($mgl^{-1}$)

NOAEL/LOAEL = No observed adverse  effect level / low observed adverse effect level

RSC = Relative source contribution (40 %)

BW = Body weight of an adult human (70 kg)

W = Daily water consumption for an adult  (2 lpd)

UF = Uncertainty factor 10 for  extrapolation from 91 day study to life  time exposure

Uncertainty factor 100 which includes 10 for extrapolation from 91 day study to life time exposure and a factor of 10 for inter individual differences in sensitivity to U toxicity

The NOAEL and LOAEL for natural U are estimated as 0.2 $mg^{-1}$ $kg^{-1}$ $day^{-1}$ (Public Health Goals 1997) and 0.06
$mg^{-1}$ $kg^{-1}$ $day^{-1}$ (Gilman 1998) respectively. In the present study LOAEL of 0.06 $mg^{-1}$ $kg^{-1}$ $day^{-1}$ is considered for
comparison of U in groundwater.  LOAEL varied from 0 $mg^{-1}$ $kg^{-1}$ $day^{-1}$ to 0.05 $mg^{-1}$ $kg^{-1}$ $day^{-1}$ which is lower
than 0.06 $mg^{-1}$ $kg^{-1}$ $day^{-1}$ in all sampling locations.  The LOAEL for uncertainty factor of 100 varied from 0 $mg^{-1}$
$kg^{-1}$ $day^{-1}$ to 0.5 $mg^{-1}$ $kg^{-1}$ $day^{-1}$. The spatial distribution of this LOAEL values less than and greater than 0.06
$mg^{-1}$ $kg^{-1}$ $day^{-1}$ is shown in Fig.5. Hence, 40% samples are considered to be under stress with an UF of 100, there
is considerable amount of threat due to consumption of groundwater in this study area with respect LOAEL
values.

**6. Conclusion**
A study was conducted to understand the levels of concentrations of U and other drinking water parameters in
the Vellore and Katpadi regions. It was observed that few samples are above the prescribed limit of uranium (30
ppb) while most of the samples fall within the range (mean = 13 ppb). Though the presence of higher uranium
concentration is not fully understood through this study but the positive correlation between U and TDS
indicated that natural U may be present due to the dissolution of these ions from rocks and the higher depth of
the wells is also a possible factor for the high presence of the Uranium. Further these results are used to derive

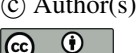



the LOAEL and NOEL values to assess the risk due to the presence of the uranium in drinking water in the
Vellore region, for which a value of 0.06 mg kg$^{-1}$ day$^{-1}$ was considered for the comparison of U in groundwater.
The LOAEL for uncertainty factor of 100 varied from 0 mg kg$^{-1}$ day$^{-1}$ to 0.5 mg kg$^{-1}$ day$^{-1}$, for which 40 % of
the samples found greater than 0.06 mg kg$^{-1}$ day$^{-1}$ in, which signifies that there is a slight risk on the public
health due to the consumption of ground water in this region.
In the current study the correlation of Uranium Concentration is restricted only with few quality parameters,
hence, by considering other physiochemical parameters and factors in further studies, the concluding
confirmation can be drawn for the proper remedial measures which may be adopted to reduce the Uranium
content in those locations in Vellore region.
**Acknowledgements**
Authors would like to thank Department of Geology, Anna University, Chennai, India for permitting to use the
geochemistry laboratory for the Uranium analysis.

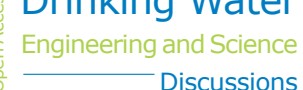

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








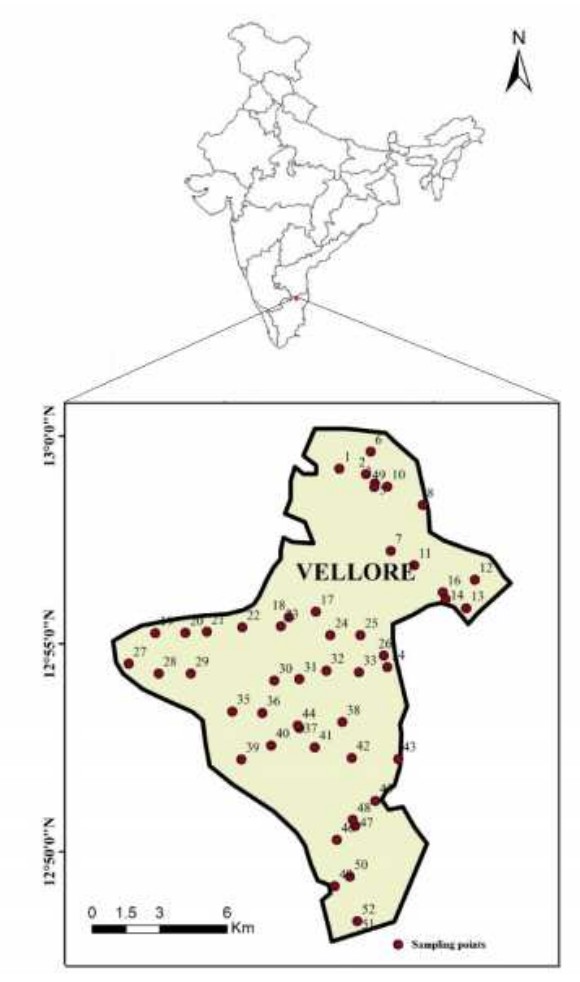



**Figure 1**: **Location map of the study area**



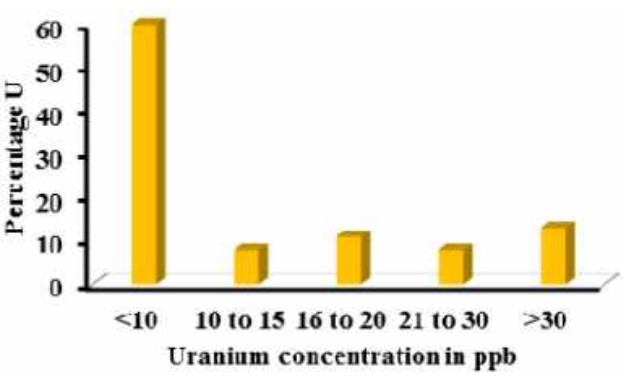

**Figure 2: Relationship between percentage of Uranium and its concentrations**


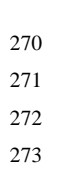


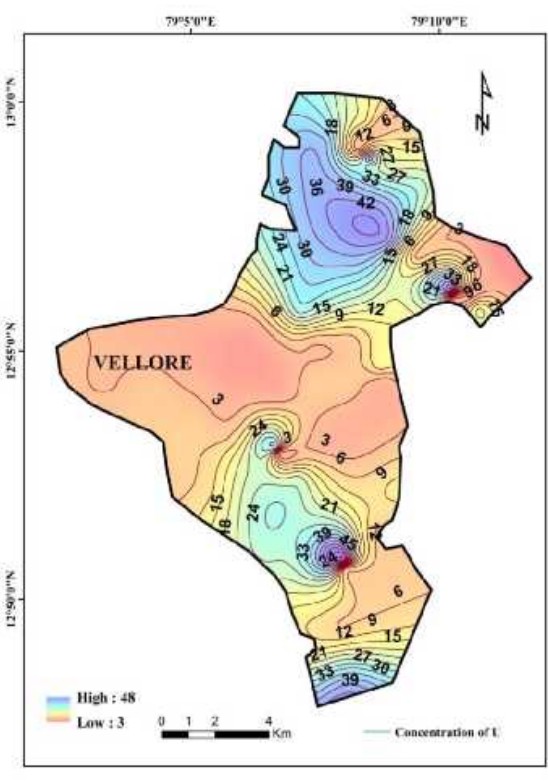

**Figure 3**: **Spatial distribution of Uranium concentration in the study area**


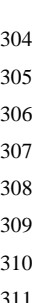









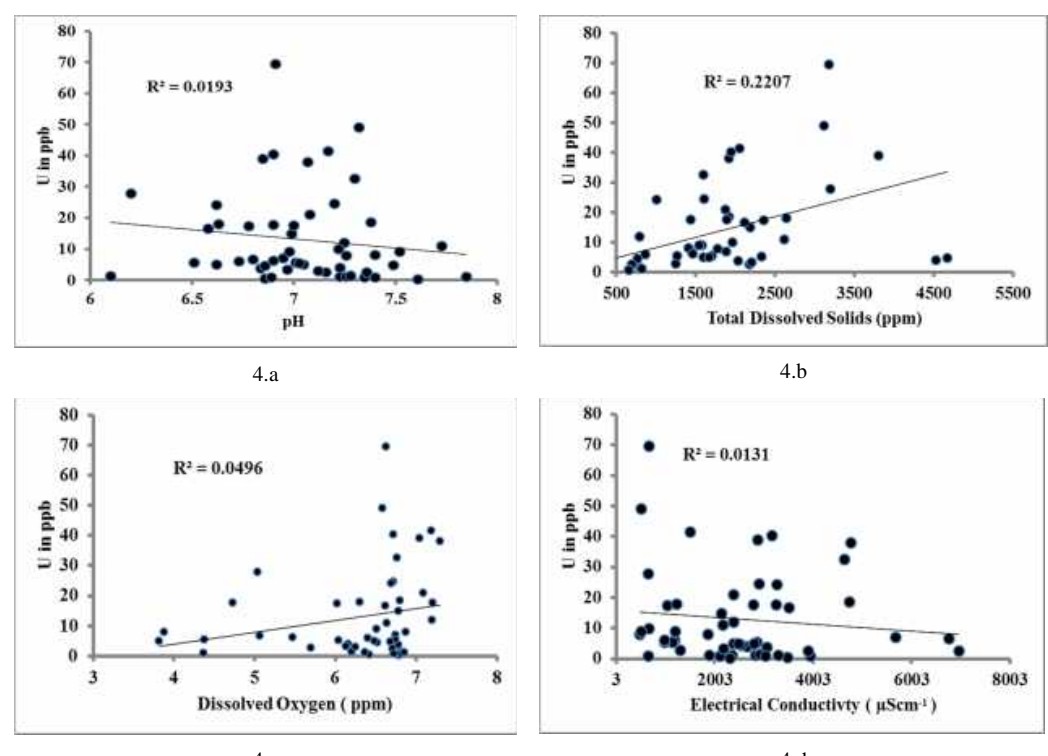

4.a          4.b

4.c          4.d

**Figure 4**: **The correlation of the Uranium concentration with pH (Fig.a),Total Dissolved Solids (**
**Fig.b),Dissolved Oxygen (Fig.c) and Electrical Conductivity (Fig.4.d)   the various parameters in drinking water**




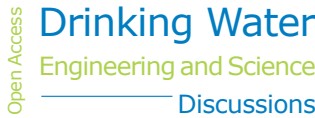


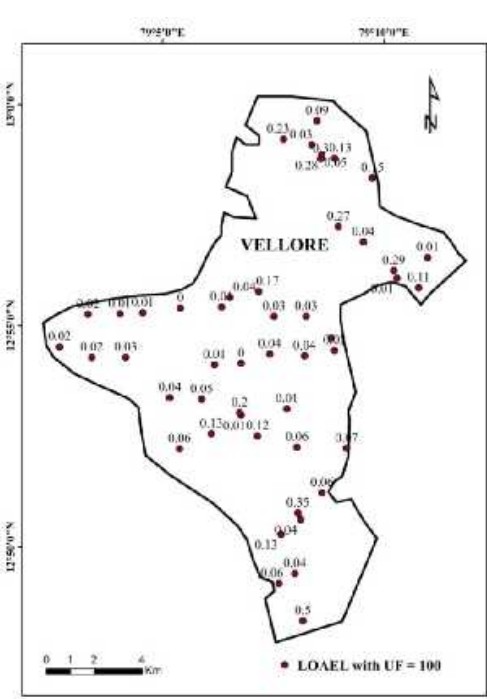

**Figure 5**: **The spatial distribution of the LOAEL**



**Table 1: Geochemistry and U concentrations of groundwater samples**

| Sample no. | pH | DO (mgl$^{-1}$) | TDS (mgl$^{-1}$) | U (µgl$^{-1}$) | Sample no. | pH | DO (mgl$^{-1}$) | TDS (mgl$^{-1}$) | U (µgl$^{-1}$) |
|---|---|---|---|---|---|---|---|---|---|
| 1 | 7.3 | 6.8 | 1923 | 18.5 | 28 | 6.9 | 6.1 | 2208 | 3.3 |
| 2 | 7.3 | 6.7 | 1604 | 32.6 | 29 | 7.1 | 6.2 | 1253 | 2.9 |
| 3 | 6.8 | 6.7 | 2040 | 3.8 | 30 | 6.8 | 6.5 | 767 | 4.5 |
| 4 | 7.2 | 6.7 | 1605 | 24.5 | 31 | 7.2 | 6.8 | 827 | 1.1 |
| 5 | 6.9 | 6.7 | 1886 | 7 | 32 | 6.8 | 6.4 | 446 | 0.4 |
| 6 | 6.8 | 7.0 | 3805 | 39 | 33 | 6.6 | 3.8 | 1596 | 5 |
| 7 | 7.2 | 7.2 | 795 | 12 | 34 | 7.0 | 6. | 1668 | 5 |
| 8 | 7.0 | 7.3 | 1925 | 38 | 35 | 7.4 | 6.7 | 700 | 2.5 |
| 9 | 7.0 | 7.0 | 1875 | 21 | 36 | 6.5 | 4.3 | 1271 | 5.5 |
| 10 | 7.1 | 7.1 | 2054 | 41.5 | 37 | 6.8 | 5.0 | 1460 | 6.6 |
| 11 | 6.9 | 7.2 | 1440 | 17.7 | 38 | 7.2 | 4.3 | 663 | 1 |
| 12 | 7.0 | 6.7 | 1710 | 5.5 | 39 | 7.2 | 6.3 | 439 | 1.2 |
| 13 | 7.4 | 6.8 | 340 | 0.9 | 40 | 7.4 | 6.8 | 1412 | 8 |
| 14 | 6.9 | 6.7 | 2190 | 14.9 | 41 | 7 | 4.7 | 1897 | 17.6 |
| 15 | 6.1 | 6.7 | 445 | 1.2 | 42 | 6.7 | 6.0 | 2355 | 17.4 |
| 16 | 6.5 | 6.6 | 2119 | 16.6 | 43 | 6.9 | 6.0 | 1582 | 9 |
| 17 | 6.9 | 6.7 | 1949 | 40.3 | 44 | 7.2 | 5.9 | 1962 | 9.9 |
| 18 | 6.6 | 6.6 | 1009 | 24.2 | 45 | 6.2 | 5.0 | 3197 | 27.9 |
| 19 | 6.7 | 6.4 | 870 | 6 | 46 | 7.2 | 3.8 | 1781 | 7.9 |
| 20 | 7.1 | 5.7 | 2183 | 2.6 | 47 | 6.6 | 6.3 | 2646 | 18 |
| 21 | 7.2 | 6.2 | 806 | 1.4 | 48 | 7.0 | 6.0 | 2333 | 5.3 |
| 22 | 7.8 | 6.8 | 328 | 1.1 | 49 | 7.3 | 6.5 | 3115 | 49 |
| 23 | 7.6 | 6.7 | 442 | 0.3 | 50 | 7.5 | 6.5 | 1549 | 9 |
| 24 | 7.3 | 6.8 | 342 | 0.8 | 51 | 6.9 | 5.4 | 1469 | 6.2 |
| 25 | 7.2 | 6.1 | 4531 | 4 | 52 | 6.9 | 6.6 | 3179 | 69.5 |
| 26 | 7.4 | 6.6 | 4671 | 4.8 | 53 | 6.8 | 6.7 | 663 | 0.8 |
| 27 | 7.7 | 6.6 | 2618 | 11 | | | | | |




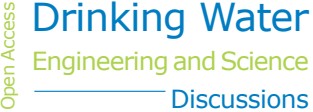










**Table 2: Statistics of the physicochemical parameters in this study**

| Parameter | pH | DO ( mgl$^{-1}$ ) | TDS ( mgl$^{-1}$ ) | U ( µgl$^{-1}$ ) |
|-----------|------|------|---------|-------|
| **Mean** | 7.02 | 6.26 | 1697.04 | 12.94 |
| **Maximum** | 7.80 | 7.30 | 4671.00 | 69.50 |
| **Minimum** | 6.10 | 3.80 | 328.00 | 0.30 |


























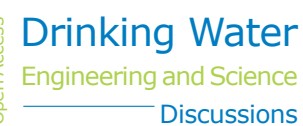



**Table 3**: **International and National Guideline values for Uranium in Drinking Water**

| Sl No | Country / Agency / Body | Limit / Guideline Value ($\mu$gl$^{-1}$ or ppb) |
|---|---|---|
| 01 | Atomic Energy Regulatory Board (AERB), India | 60 * |
| 02 | World Health Organisation (WHO, 2011) | 30 ** |
| 03 | United States Environmental Protection Agency (USEPA) | 30 |
| 04 | Germany (2001) | 10 |
| 05 | Russian Federation | 15 |
| 06 | Canada | 20 |
| 07 | Switzerland | 30 |

**(* Radiological toxicity, ** Chemical toxicity)**

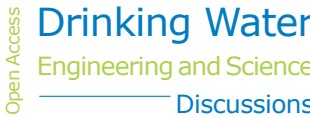


**Table 4**: **Co efficient of correlation of the Uranium concentration with the various parameters in drinking water**

| Parameters | pH | DO | Methyl alkalinity | Total Dissolved Solids |
|---|---|---|---|---|
| Co efficient of Correlation | 0.0193 | 0.0496 | 0.059 | 0.2207 |
| Correlation Type | No correlation | No correlation | No correlation | weak positive correlation |