# Peer review of "Assessment of Uranium concentration in groundwater and its"

_Drinking Water Engineering and Science, 2019_

## Referee Comment (RC1) · Anonymous Referee #1 · 2 Apr 2019

General comments: The authors describe uranium concentrations in groundwater of a city in Tamil Nadu, India. They have analysed water samples for EC, pH, DO, alkalinity, hardness and uranium concentration 53 locations and explore their suitability for drinking purpose. Unfortunately, the information given in the manuscript is inadequate for a reader to examine their interpretations and to follow their conclusions. Though the general structure of the manuscript is acceptably given, it is not easily readable and confusing, as the authors do not provide sufficient clarification of the data. The manuscript reserves vast scope for restructure and addition of new information by in-between analysis. The methodology is incomplete. Key hydrochemical processes that govern uranium in groundwater such as the ORP has not been taken into consider-

ation. Aquifer characteristics, local groundwater flow etc. has not been considered in evaluating the results. Discussion of the results is lacking. Many figures and tables have overlapping information. The write-up, tables and figures can be better organized.

Specific comments: - The abstract is not comprehensive. It does not report any results of the study, i.e. the range of measured uranium concentrations. The study involves only a few parameters and could be listed in the abstract instead of mentioning as laboratory and insitu parameters.

Introduction - L28-29, needs a reference - L30, does the authors mean 'groundwater'? - What is referred to as 'drinking water'. It is confusing as the authors use groundwater (GW), surface water (SW) and drinking water (DW) one after another, without linking them. Does the authors mean uranium occurs naturally and through anthropogenic sources in GW and SW and this is the source for DW. - L33, quoting a single reference seems unfitting. Sure several studies have come to this inference. - L33-34 'Mainly U is present in groundwater rather than surface water' needs a reference to support this statement - L35, please avoid vague information such as some parts of the body. Be specific. - L39, what is meant by 'test area'? - The introduction has huge scope for re-structure. In many developed nations, GW/ SW used as a source of DW is treated and tested for all important parameters to be within the DW limits before public supply. The authors should clearly mention that in developing nations like India, groundwater which may be contaminated by naturally occurring uranium is used directly for drinking use. Without this, international readers will not be able to understand the current situation in the study area. - Several references are introduced in the introduction, but the position, novelty and aim of the work among the related studies is not explained. - The authors mention few studies from the USA in the introduction followed by studies from India. The literature review is restricted to selected geographical regions and omits an overall view of the status in the world. This should be revised. - Just reporting uranium concentrations from different parts of India may not be useful for the readers. Authors should link these studies and also mention if there are health issues in these areas as

they had earlier specified in the introduction. - A recent large-scale uranium study in India (DOI: 10.1021/acs.estlett.8b00215) has not been included in the introduction by the authors. As a reader, I feel that a comprehensive assessment on the status of uranium in GW in India (as put-forth in the introduction) has not been made. - L58-59, this statement needs reference. - L61, it is stated that it is important to provide clean and safe DW to the people. This includes not only uranium but also various other chemical constituents of DW. In that case, it should be clarified how only uranium was chosen for this study.

Study area - It should be made clear if the information on the rainfall, relative humidity, soil type, geology etc were primary data collected by the authors or secondary data from other sources. - Mention the area of the study site. - L71, what is the depth of the open and bore wells?

Methodology - Methodology is poorly presented. - What is the detection limit for uranium in laser fluorimeter? - More information about the sampling is needed. Was the samples filtered prior to collection? Was any measures taken to stabilize the samples before taking them for laboratory analysis? At what depth were the samples collected? Was the groundwater level measured? How many samples were collected? - Redox processes play a key role in the occurrence of uranium. The authors should mention about this.

Section 4- R&D - The authors have not mentioned measuring TDS in section 3.1. However, this is discussed in section 4. Please clarify. - L92, what is SEPA? - TDS and EC are directly proportional. Hence, it is logical to expect high TDS with high EC. Such statements should be avoided. - L93-94, not all hard rock regions should have high TDS and EC. Rephrase this sentence. - Table 1 lists the composition of all the measured parameters from all samples. Hence, a separate table for the statistical summary is unnecessary. This can be included within table 1. - Authors mention that certain percentage of samples exceed the limit for some parameters. Reference is required for all these limits eg. 500 mg/l TDS in L95 - L98 what is meant by percentage

U? does the authors mean % of samples within varied range of uranium? - L98-100 what does the authors try to infer by classifying the samples based on different range of uranium? - L102-103 rephrase for better clarity - L 103, What is AERB? Authors should expand the abbreviations of international organisations at the first mention. Readers will not be able to understand these abbreviations. What is the standard proposed in India? - L106 delete (i.e. >30ppb). This has been mentioned in the first part of the same sentence. - L107-108, the sources for the occurrence of uranium in the northern part is mentioned as charnokite, gneiss and granite. If so, what is the geology in the other parts of the study area where uranium concentrations are low. Since the source of uranium is attributed to the geology of the region, a geology map of the study area should be included. Also, please provide the relative depths of each of the formations in the study area. - Does depth of the sample collection have a role in the variation in uranium concentration? Is it possible to compare the water levels with the uranium concentration in the area? - L110, a correlation of 0.6 and above is considered as a good correlation. 0.2 cannot be considered as a slight positive correlation. - L108-109 and L110-111 are sentences with the same meaning. Such cases occur often in the manuscript. - L114- what are the other factors? Please be specific. - A clear analysis and interpretation about the role of the redox processes on the uranium concentration in groundwater is lacking. - Land use also plays a major role in uranium concentration in GW. This aspect should be discussed. Land use information should also be included in the study area. - Section 5 - Section 5, isn't this also a part of the R&D? - L116-120, these are general information which may be restricted to the introduction. Should be moved from R&D. - References are required for the values mentioned in the equation for NOAEL and LOAEL. How was the average weight and water consumption etc. arrived at? The equation should be explained briefly. What is the value of C? - Calculated values for LOAEL are mentioned here, what about NOAEL? - There is very little information in the discussion or conclusions that would put the results of this study in a broader context. The statistics presented is limited to this site, which may not be applicable to other areas because of different geological

formation. A valid discussion which integrates the current study with the peer-reviewed literature is therefore required to ensure it is of wider applicability.

Figures and tables - Show Tamil Nadu in figure 1 - Figure 2, it is not understandable what percentage U means. - Figure 3, wouldn't it be easier to understand this figure, if the range of the contours indicates <30, 30-60and >60 ppb? - Figure 4, title should be rephrased - Isn't figure 4 and table 4 the same information - Figure 4 As mentioned earlier, TDS and EC cannot have varied correlations with uranium. Please check the data and the graphs.

Technical corrections: - Please proofread the manuscript to avoid any unexpected typos or grammatical errors. - L17-18 Samples were collected and analysed for various water quality parameters in the laboratory and as well as insitu. - Authors should check the entire manuscript for the inappropriate use of upper case letters in the middle of a sentence eg. L 20. - Keywords are very few, could be increased to 5-6 - Please be consistent with using the subscripts and superscripts - Consistent use of U for uranium should be adopted, eg L31, 89 - Consistent use of units: uM, Ug and ppb. Follow uniformity. - Groundwater or ground water? - Please check the references list. Eg L 47 it is mentioned as Brindha et al. But in the reference, it is Brindha and Elango. - Figures contain overlapping information. Figures with similar information should be combined. - All figure quality should be improved.

---

## Referee Comment (RC2) · Anonymous Referee #2 · 3 Apr 2019

General comments: - It is not clear how this research relates to existing literature - Ground water = groundwater - Once "U" is introduced, always use "U" - Be consequent in units: or ppb or $\mu$M or $\mu$g/L - Use "concentration" instead of "amount", "quantity" or "content" - The section 5.1 is not relevant for the manuscript since this is the basis for the WHO and USEPA guidelines. Specific comments: - 14, what is meant with "major" and "minor" ions? - 15, "has been" - 17, delete "the" and "and" - 19' 'is exceeded' - 20, delete "for" - 27, "minute quantities" = "Low concentrations" - 29, "in" = "on" - 29-30, give references - 30, what is meant by "significant", "quantities" = "concentrations" - 31, "processes", "mobilizes" - 33, "U is mainly present" - 34, "amount" = "concentration" - 35, "Higher concentrations of U can affect" - 36, delete "and it is" - 44, "water due to U
was" - 49, delete "the study of" - 53, delete "the" - 54, "a concentration range of" - 58, "boreholes"(check also rest of the manuscript), are these boreholes with hand pumps or are the hand pumps placed on "dug wells"? - 59-60, not clear what is meant, why is this relevant (and to do with the U concentration)? - 62, explain why. - 65, is Vellore a "region" or a "city"? - 70-71, see comment on line 59-60 - 73, repetition and not clear what is the relation with U. - 77, "the study area" - 79, not clear what the radius of 5 km has to dot with the grids of 2 by 2 km. - 82, "the" = "a" - 83-84, give reference - 85-86, not clear what is meant - 89, "the different locations" - 90, "in Table 1" - 90-91, rephrase - 91, delete "in the present study" - 92, delete "very" and explain what is meant by "high" - 93, "Similar to TDS, the values for EC are high and ranged. . ." - 93-94, unclear what is meant - 94, "the groundwater samples exceeded . . ." The value of 500 mg/l refers to? - 95, "are lies"? - 95-97, "Whereas the DO varied from 3.8 mg/l to 7.3, mg/l." (delete the rest of the sentence) - 97, combine Table 1 and 2. - 98, what is meant by "percentage of U and its concentration" - 99, "the samples in the study area had an U concentration" - 100, "are" = "were" - 103, delete "as per". What is AERB? - 103-104, "thus only one location has crossed this limit". - 104, delete "as per", "have permissible limits of" - 105, "is" = "of" - 106, "exceeded this limit" (delete i.e. >30 ppb). "The spatial distribution" - 107-109, explain how this is known - 109, "not a strong correlation" - 110, explain what can be the positive correlation between U and TDS and if there are references. - 111-114, delete sentences, no extra information. - 116-118, delete sentences - 118-120, is for introduction and give also references - 141-142, not clear why this is and how this can be explained (with references) - 143-151, not relevant for this manuscript.

---

## Referee Comment (RC3) · Anonymous Referee #3 · 14 Apr 2019

The authors in this paper monitored the levels of U in drinking water of area in Vellore, a city in the Northern part of the Tamilnadu, India. Some parameters such as EC, pH, DO, alkalinity, and hardness were measured also for 53 locations. Unfortunately, this paper could not publish in Drink. Water Eng. Sci. journal in the current quality due to the following comments: (1) In the Abstract, only U was mentioned and the rest of the monitored parameters are ignored. The obtained results are not existed in the Abstract such as U concentrations. (2) In the introduction: # Line 33, the U concentrations should be taken into consideration in surface water, is there is any reference exclude that? # Paragraph (Lines 58-62), the groundwater (GW) is used in this area without any treatment?. Authors did not mentioned the situation of water treatment facilities. # Line

61, the aim of work is only to find the U concentration in the GW?, What about the rest of parameters. Authors did not explain why the measured parameters are sufficient for governing the quality of drinking water. # There is a lack of novelty and the aim of this study is not linked to the international community. (3) In Study area, References for the data of humidity and soil types are missed. (4) In the Methodology: # What is the standard methods reference used for the investigation of alkalinity, hardness, and U?. There is no validation for the method of U detection? # The data for samples collections and the treatment before analysis is missed. # The method for measuring TDS is missed, however TDS results are exist in the Results and Discussion. (5) In the Results and Discussion: # Table 1 and Table 2 can be merged in one table. # Line 93 reference is needed for this sentence "hard rocks resulted in high TDS and EC values" # The importance of Fig. 2 "the relationship between percentage U and its concentration" was not clarified. # What is SEPA in Line 92 and AERB in Line 103. # There is no explanation for the presence of high and low U concentrations in the same area. # Line 110, there is no correlation between U and TDS with R2= 0.2. # Line 114, what is the other factors that explain the presence of U?. # There is no full picture for the area under study to describe the obtained results such as human activities, depth of wells, etc. # The equation is for determination the value of C and not for the determination of NOAEL and LOAEL. This equation should be clarified. (6) Authors did not recommend solutions for the problem of high U concentration in some samples and conclusion is limited without comparison with literature.